# The Effect of Exogenous Application of Quercetin Derivative Solutions on the Course of Physiological and Biochemical Processes in Wheat Seedlings

**DOI:** 10.3390/ijms22136882

**Published:** 2021-06-26

**Authors:** Marta Jańczak-Pieniążek, Dagmara Migut, Tomasz Piechowiak, Jan Buczek, Maciej Balawejder

**Affiliations:** 1Department of Crop Production, University of Rzeszow, Zelwerowicza 4, 35-601 Rzeszow, Poland; dmigut@ur.edu.pl (D.M.); jbuczek@ur.edu.pl (J.B.); 2Department of Food Chemistry and Toxicology, University of Rzeszow, Ćwiklińskiej 1A, 35-601 Rzeszów, Poland; tpiechowiak@ur.edu.pl (T.P.); maciejb@ur.edu.pl (M.B.)

**Keywords:** flavonoids, quercetin derivative, wheat (*Triticum aestivum* L.), antioxidant activity, chlorophyll content, chlorophyll fluorescence, gas exchange

## Abstract

Quercetin, classified as a flavonoid, is a strong antioxidant that plays a significant role in the regulation of physiological processes in plants, which is particularly important in the case of biotic and abiotic stresses. The study investigated the effect of the use of potassium quercetin solutions in various concentrations (0.5%, 1.0%, 3.0% and 5.0%) on the physiological and biochemical properties of wheat seedlings. A pot experiment was carried out in order to determine the most beneficial dose of this flavonoid acting as a bio-stimulant for wheat plants. Spraying with quercetin derivative solutions was performed twice, and physiological measurements (chlorophyll content and fluorescence as well as gas exchange) were carried out on the first and seventh days after each application. The total phenolic compounds content and the total antioxidant capacity were also determined. It was shown that the concentrations of potassium quercetin applied have a stimulating effect on the course of physiological processes. In the case of most of the tested physiological parameters (chlorophyll content and fluorescence and gas exchange) and the total antioxidant capacity, no significant differences were observed in their increase as a result of application with concentrations of 3.0 and 5.0%. Therefore, the beneficial effect of quercetin on the analysed parameters is already observed when spraying with a concentration of 3.0%.

## 1. Introduction

Common wheat (*Triticum aestivum* L.) is the most widely cultivated species in the world with the sown area of 215.9 million ha [1]. Wheat represents the main source of calories and protein for the world’s population. This contributes to the high nutritional importance of wheat proteins, especially in regions, where wheat products constitute the significant part of the human diet. Therefore, at present, food security largely depends on the increased production of cereals, in particular wheat [2,3,4]. During cultivation, wheat is exposed to both biotic and abiotic stress factors, which damage plants and contribute to inhibition of their growth and reduction of grain yield [5]. This situation leads to the development of oxidative stress due to overproduction and accumulation of highly toxic reactive oxygen species (ROS) (O_2_**·**^-−^, H_2_O_2_, ^1^O_2_, HO_2_**·**^−^, OH**·**, ROOH, ROO**·** and RO**·**), most often as a result of an imbalance between production of ROS and the activity of scavengers [6,7,8]. As a result of oxidative stress, cell organelles and cell membranes are damaged, which can eventually lead to cell death. There are protective mechanisms in plants against oxidative stress, leading to the reduction of ROS production and to their scavenging [5]. The antioxidant system in plants is composed of enzymatic (superoxide dismutase, ascorbate peroxidase, guaiacol peroxidase, catalase, etc.) and non-enzymatic antioxidants (flavonoids, carotenoids, ascorbate, glutathione, tocopherols, etc.) [9]. Flavonoids belong to the group of low molecular weight antioxidants produced in plants. They constitute a secondary antioxidant system because they are activated under conditions of severe stress when the activity of antioxidant enzymes is exhausted. Phenolic compounds and flavonoids play an important physiological role in the protection of plants exposed to stress [10,11] and they are essential for the defence mechanisms of plants, as they ensure their specific adaptation to changing environmental conditions [12]. Their role as scavengers of ROS is to locate and neutralise free radicals before they cause cell damage, making them important compounds for plants exposed to adverse environmental conditions [13].

Quercetin (3,3′,4′, 5,7- pentahydroxyflavone) is one of the most abundant flavonoids among plants belonging to the class of flavonols. Quercetin, classified as a phenolic compound, has a strong antioxidant effect as it helps maintain oxidative balance [14]. This flavonoid is located in the chloroplast envelope, probably in the outer envelope membrane [10]. This place of occurrence suggests its role in regulating and limiting the intensity of light available to the plant [15]. Quercetin plays an important role in the process of protecting plants against the effects of stress such as: UV radiation or osmotic stress, as has been confirmed in numerous studies [16,17,18]. In osmotic stress, glycosidic derivatives of flavonoids play a significant role in the osmoregulation process. Glycosylation most often takes place in the cytoplasm, while the deposition of these substances occurs in the vacuoles. Accordingly, the glycosides are spatially separated and functionally from chloroplasts [19].

Flavonoids, including quercetin, have a considerable ability to change the polar auxin transport and signalling by modifying a large number of proteins involved in this process. Thus, they are auxin transport regulators and can therefore be considered responsible for most of the developmental patterns in plants [20].

Widely used plant protection products are applied in order to increase the efficiency of agricultural production. However, they have a significant impact on the natural environment and human health. Therefore, more and more emphasis has being placed on the use of safe agents—biostimulants, which could contain flavonoids—including quercetin [14,21]. However, despite the large number of publications on the impact of external application of phenolic substances on the physiological processes occurring in plants [22,23,24], information about it in wheat seedlings is insufficient [25]. The aim of the study was to evaluate the photosynthetic apparatus efficiency and antioxidant properties of wheat seedlings as a result of exogenous application of various concentrations of quercetin. The obtained results will allow future studies to assess the usefulness of this flavonoid as a preventive agent protecting plants against biotic and abiotic stresses.

## 2. Results

### 2.1. Basic Characteristics of the Potassium Quercetin Derivative

As part of the study, the optimal molar fractions of the reagents in which the potassium quercetin derivative formation reaction proceeds efficiently and all the functional groups that participate in the reaction were first determined. In preliminary tests it was noticed that the addition of KOH solution to the methanolic quercetin solution causes a significant change in the colour of the solution, which was observed in the form of changes in the absorbance of the solutions (Figure 1). Therefore, Job*’*s method was used to determine the optimal molar fractions of the reagents. Based on the Job curve, it was found that the optimal molar fraction of quercetin to potassium hydroxide should be 1:7.

The quercetin derivative was subjected to UV-vis analysis (Figure 2). On the basis of the UV-Vis spectra, shifts in the position of the absorption maxima for quercetin complexes towards long wavelengths (bathochromic shifts) and a decrease in the absorption intensity (hypsochromic effect) in relation to the quercetin standard were noticed, which indicates the formation of derivatives.

The potassium quercetin derivative was then analysed for anti-radical activity against ABTS and DPPH. In the course of the study, it was found that the antioxidant activity against ABTS and DPPH was significantly lower than the activity of the quercetin standard (Table 1). However, the prepared potassium derivative showed a higher ABTS and DPPH scavenging ability than ascorbic acid and comparable to Trolox.

### 2.2. Relative Chlorophyll Content

The relative chlorophyll content in wheat leaves treated with quercetin derivative solutions grew with the increase of the concentrations applied (Figure 3). No difference was found in the chlorophyll content between the concentrations of quercetin 3.0% and 5.0%, except for the first measurement (Term 1), where spraying with a concentration of 5.0% resulted in a significantly higher value of the tested parameter by 8.5% compared to spraying with a concentration of 3.0%.

The relative chlorophyll content grew by subsequent measurement dates. The increase in chlorophyll content was observed the day after the first and second application (Terms 1 and 3) with 1.0% quercetin solution, while in the case of the application of a concentration of 0.5%, the increase in chlorophyll content was found only the day after the second spraying (Term 3). As a result of the application of 5.0% quercetin derivative solution, the growth in chlorophyll content was noticed only after the first and seventh day after the application of the second spraying (Term 3 and 4). The greatest changes in the content of chlorophyll were recorded with the application of a concentration of 3.0%, the use of which resulted in a significant increase in the value of the tested parameter on subsequent measurement dates.

### 2.3. Chlorophyll Fluorescence

Wheat plants reacted to foliar spraying with quercetin derivative solutions by increasing the values of chlorophyll fluorescence parameters (Figure 4a–c). The stimulating effect of this flavonoid as compared to the control was found for all parameters on each measurement date, except for the PI indicator, for which in Term 1 no difference was found between the control and the 0.5% solution spray.

For the parameters F_v_/F_m_ and F_v_/F_0_, no differences were observed in the use of concentrations of 0.5% and 1.0% and 3.0% and 5.0% on both the first and the seventh day after the first application of quercetin derivative (Term 1 and 2). After the second spraying on the first day (Term 3), the value of the F_v_/F_m_ parameter increased significantly after the application of the concentrations of 3.0% and 5.0% compared to the concentration of 0.5%, while on the seventh day (Term 4), no differences were observed between the concentrations of 3.0% and 5.0%. After the second application of quercetin derivative, a significant increase in the F_v_/F_0_ value was found as a result of the application of higher concentrations the day after the treatment, while on the seventh day after the treatment an increase was also observed, except for the concentrations of 1.0% and 3.0%, in which the F_v_/F_0_ values did not differ. After the first application, the PI value did not differ between the concentrations of 1.0% and 3.0%, while after the second spraying (Term 3 and 4), an increase in the PI value proportional to the quercetin derivative concentrations applied was observed.

The value of chlorophyll fluorescence parameters increased by the subsequent measurement dates. Only in the case of Fv/Fm, no significant differences were observed, except for the application of the concentration of 1.0%, which caused an increase of 2.6% after the second application (Term 3 and 4). Significant increases in F_v_/F_0_ and PI values were already observed on the seventh day after the first application (Term 2), with the exception of the application of a concentration of 1%, which caused an increase in F_v_/F_0_ only after the application of the second spraying.

### 2.4. Gas Exchange

An effect of the application of quercetin derivative solutions on an increase in the values of parameters (g_s_, P_N_ and E) (Figure 5a–c) and a decrease in C_i_ (Figure 5d) compared to the control was found. On all measurement dates, a significant increase was found in the values of E and g_s_ parameters proportional to the quercetin derivative concentrations applied. Only on the seventh day after the first application (Term 2), there were no significant differences in the value of E as a result of spraying with 3.0 and 5.0% solutions. Quercetin derivative concentrations 0.5 and 1.0% as well as 3.0 and 5.0% did not significantly influence the differentiation of the P_N_ value in Term 2, 3 and 4. In Term 1, the P_N_ value did not differ between the concentrations of 0.5 and 1.0%, while as a result of spraying with 3.0% and 5.0% concentrations a significant increase in P_N_ was observed. A significant decrease in the value of C_i_ was observed with the application of higher concentrations of quercetin derivative (Terms 1, 2 and 3). In term 4, the highest value of C_i_ (at the control level) was demonstrated by using concentrations of 0.5 and 1.0%.

The values of the gas exchange parameters increased with the successive dates of measurements. However, no significant changes were found for the parameters: C_i_ (5.0%) and g_s_ (0.5, 3.0 and 5.0%). The stimulating effect of quercetin derivative was shown only after the second application, with the exception of the P_N_ (3.0 and 5.0%) and E (3.0%) parameters, the value of which increased significantly after the first application (Term 2).

### 2.5. Total Antioxidant Capacity

Figure 6 shows the effect of quercetin derivative spraying on total antioxidant capacity. As a result of its use, the stimulating effect of this flavonoid was demonstrated. The use of a 0.5%concentration resulted in an increase in on total antioxidant capacity by 27.4% compared to the control. A further increase in the value of this parameter was visible only after spraying with a 3.0% quercetin derivative solution. As a result of spraying with the concentrations of 0.5% and 1.0% as well as 3.0% and 5.0%, no significant differences in the increase of total antioxidant capacity were found.

### 2.6. Total Phenolic Compounds

Figure 7 shows the total amount of phenolic compounds, which increased in proportion to the quercetin derivative concentrations applied. The differences between particular concentrations had a significant effect on the indicator tested. As a result of treatment of the plants with a concentration of 5.0%, the amount of phenolic compounds increased by 32.8% compared to the control. The total polyphenol content has been shown to be linearly dependent on the dose of the quercetin derivative. The coefficient of determination R^2^ is characterized by a very good fit and amounts to R^2^ = 0.99.

## 3. Discussion

Plants, due to their sedentary lifestyle, are more exposed than animals to damage caused by environmental pressure (e.g.**,** stress related to UV radiation, heavy metals, salinity, drought or temperature changes). This has caused them to evolve various mechanisms that allow them to adapt to unfavourable conditions [26]. As a result of exposure to stress factors, many flavonoid biosynthesis genes, including quercetin, are induced, which play an important role in plant life by regulating physiological processes [27]. However, a large role is played by their antioxidant properties, which are activated by the occurrence of environmental stresses and attack from pathogens. The process of flavonoid oxidation is catalysed by polyphenol oxidases and peroxidases [28]. Studies by Bieza and Lois [29] on *Arabidopsis thaliana* L. mutants that are unable to synthesise flavonoids, which are damaged by light stress, proved the antioxidant properties of these compounds through the absorption of UV rays.

Taking into account the limitations of the biological potential of cultivars, it seems appropriate to additionally search for substances that can reduce environmental stress, which will result in an increase in yield. Such substances, called bio-stimulants, are natural growth regulators that contribute to an increase in the physiological activity of plants through protein synthesis. So, quercetin, which is a strong antioxidant, provides plants with tolerance to biotic and abiotic stresses [26,30]. The quercetin molecule undergoes redox changes (i.e.**,** electron transfer or hydrogen donation reactions). Oxidation of quercetin is widely regarded as the process leading to the loss of antioxidant properties. However, research conducted by Fuentes et al. [31] prove the greater power of the antioxidant properties of quercetin oxidation products, especially benzofuranone, compared to quercetin. The reduction of oxidative stress caused by benzofuranone in cell model systems has been confirmed in studies performed by Fuentes et al. [32]. Secondary metabolites may also be valuable in the breeding of varieties with features of resistance to unfavourable environmental conditions, which increases their role in agriculture [33].

In our own study on the effect of exogenous application of quercetin derivative solutions on wheat seedlings, the stimulating effect of this compound on the course of physiological processes was found. As a result of the spraying, a positive effect of quercetin on the value of the tested physiological parameters (chlorophyll content, values of selected parameters of chlorophyll fluorescence and gas exchange), proportional to the applied concentrations, was demonstrated.

The stimulating effect of exogenous application of quercetin derivative on the content of chlorophyll has also been confirmed in studies conducted on tomato seedlings by Parvin et al. [17]. As in the authors**’** own study, an increase in quercetin concentration resulted in an increase in chlorophyll content in leaves. Additionally, in studies on corn seedlings Araniti et al. [34] showed an increase in the content of photosynthetic pigments, including chlorophyll, under the influence of a potent phenolic compound—cinnamic acid.

The stress to which plants are exposed negatively affects photosynthetic electron transport in PSI and PSII and the biosynthesis of chlorophyll [35]. The photosynthetic apparatus is therefore a sensor of various environmental stresses that may be responsible for an imbalance in cellular energy due to the modification of the redox status [36]. Quercetin plays a role in the repair processes of the photosynthetic apparatus leading to an increase in photosynthetic efficiency, which was confirmed in studies by Ma et al. [37], where the F_v_/F_m_ chlorophyll fluorescence parameter increased due to external application of quercetin. These results are consistent with those obtained in studies by Meng et al. [38], in which increased accumulation of flavonoids eliminated photoinhibition and damage to the membranes of the photosynthetic apparatus, thereby increasing the efficiency of the photosynthetic process.

Quercetin participates in the light-dependent phase of photosynthesis, during which it improves electron transport [39]. Application of quercetin may alter photosynthetic properties due to structural changes in the thylakoid membranes which lead to a greater degree of thylakoid light-scattering. Changes in the thylakoid membranes caused by the addition of quercetin affect the transfer of electrons between the pigment-protein complex. Structural changes caused by increased cell membrane fluidity may lead to increased transfer energy from PSII to PSI, which improves the process of photosynthesis [16]. Flavonoids, including quercetin, play a significant role in changing the properties of cell membranes, which confirms their antimicrobial nature [38].

In a study by Yildiztugay et al. [40] on *Phaseolus vulgaris* L. plants, it was also shown that the application of the flavonoid—naringenin had a positive effect on the course of the photosynthesis process, which resulted in an increase in the parameters of chlorophyll fluorescence. As a result of an increase in naringenin concentration, a significant increase in the F_v_/F_0_ value was observed, but there was no increase in the F_v_/F_m_ value compared to the control [41]. In our own experiment, all the tested parameters of chlorophyll fluorescence (F_v_/F_m_, F_v_/F_0_ and PI) showed a significant increase in value compared to the control, along with an increase in the applied concentrations. These discrepancies could be due to differences in the action of quercetin and naringenin.

Environmental stresses limit the process of photosynthesis mainly by reducing stomatal conductivity, causing oxidative stress. As a result, the activity of Ribulose-1,5-biphosphate carboxylase/oxygenase (RUBISCO), which is responsible for the fixation of CO_2_ in the dark phase of photosynthesis, decreases [42]. In own study, as a result of the application of quercetin derivative, an increase in the values of gas exchange parameters (g_s_, E and P_N_) was found. A similar relationship was found by Yildiztugay et al. [41] due to the application of naringenin. The increase in g_s_ was a consequence of the involvement of quercetin in stomata opening by strongly influencing the signalling pathways leading to the closure of the stomata regulated by abscisic acid (ABA) [43]. The stomata contain anion channels located in the cell membrane and are activated by a variety of stimulators such as ABA, NADPH metabolism, and membrane anion flow via voltage-gated K^+^ transport channels. As a result of environmental stresses, the amount of ROS is increased which causes stomata closure in order to minimise water loss [44,45]. Flavonoids, in particular quercetin derivatives, exert a profound effect on ABA signalling by antagonising ABA-induced stomatal closure. Watkins et al. [46] presented a model of the operation of flavonoid-induced stomata opening. As a result of stress, an ABA-induced respiratory burst oxidase occurs, which causes a burst of ROS, leading to the closure of the stomata, which in turn contributes to a decrease in transpiration.

Our own study also showed an increase in the total antioxidant capacity and the total amount of phenolic compounds as a result of quercetin derivative application. In plants growing under stress conditions, increased accumulation of phenolic compounds leads to greater plant tolerance. The increase in plant resistance is related to various functions of the polyphenols mainly based on their ability to capture ROS. Abiotic stresses also activate cell signalling processes leading to the transcriptional upregulation of the phenylpropanoid pathway [42]. The biosynthesis of phenolic compounds is related to the activation of key biosynthetic enzymes and the upregulation of the key genes of the phenylpropanoid branch—including the phenylalanine ammonia-lyase enzyme (PAL, EC 4.3.1.5), which is involved in the biosynthesis of antioxidant phenolic compounds [47]. The precursors of these compounds are aromatic amino acids generated in shikimic acid pathway. They include phenylalanine (Phe), which is a substrate for many secondary metabolites [23,48]. Studies conducted by Feduraev et al. [49] showed that plants growing on a medium enriched with this amino acid enhanced the expression of all selected genes related to flavonoid biosynthesis. Phe is initially deaminated by PAL to cinnamic acid. This acid is then converted into cinnamoyl-CoA by 4-coumaroyl: coenzyme A ligase (4CL, EC 6.2.1.12) or converted to p-coumaric acid by a P450 cytochrome monoxygenase enzyme, cinnamate-4-hydroxylase (C4H EC 1.14.13.11). Then p-coumaric acid is converted to phenolic CoA thioesters catalysed by 4CL by attaching CoA to a phenolic compound, producing p-coumaroyl-CoA. Phe provides 6-carbon ring and 3-carbon side chain that is central to all phenylpropanoids and then enters the flavonoid pathway to generate major groups of flavonoids including flavonols [48]. Quercetin derivatives also have the ability to form complexes with Cu and Fe ions, contributing to the inhibition of ROS production in the Fenton reaction and may protect chloroplasts against singlet oxygen produced by visible light [50,51].

## 4. Materials and Methods

### 4.1. Pot Experimental Design

The pot experiment was carried out at the University of Rzeszów (Poland). In plastic pots (11 × 11 cm, 3 kg soil/pot) soil with a clay sand particle size composition [52] and slightly acidic pH (KCl pH 6.35; H_2_O 6.52) was placed. Four seeds of Artist cultivar winter wheat (breeder Deutsche Saatveredelung AG, Lippstadt, Germany) were sown in each pot. The experiment was carried out in 4 repetitions in a growth chamber (Model GC-300/1000, JEIO Tech Co., Ltd., Seoul, Korea) at a temperature of 22 ± 2 °C, humidity 60 ± 3% RH, photoperiod 16/8 h (L/D) and a maximum light intensity of about 300 µmol m^−2^ s^−1^. During the experiment, the soil moisture in the pots was kept constant at the level of 60% of the maximum water holding capacity (WHC). The pot settings were changed at random every 5 days. Potassium quercetin solutions were applied in the following concentrations: 0.5%, 1%, 3% and 5%. The quercetin derivative was diluted in deionised water (20 mL of solution per pot). Two spraying treatments were performed: 10 and 17 days after the 14 BBCH stage with a hand sprayer. The BBCH scale (Biologische Bundesanstalt, Bundessortenamtund CHemische Industrie) was given according to Meier [53].

Determination of physiological parameters (chlorophyll content and fluorescence as well as gas exchange) was carried out four times on the first fully developed leaves of wheat: on the first and seventh days after each spraying.

### 4.2. Synthesis and Physicochemical Properties of the Potassium Quercetin Derivative

#### 4.2.1. Determining the Optimal Stoichiometric Ratios between the Reagents

Job’s method [54] was used to determine the optimal ratios between the reagents, i.e., quercetin and potassium ions. To this end, a methanolic solution of quercetin (Sigma-Aldrich, Steinheim, Germany) at a concentration of 1 mM was added to a 1 mM KOH solution in volume ratios of 1:20-10:1, after which the absorbance was measured at the wavelength λ = 374 nm. Graphs of the relationship between the molar fraction of the reagents and the absorbance of the solutions were plotted, and then, using the first derivative method, the molar fraction of the reagents in which the absorbance value was minimal was determined.

#### 4.2.2. Synthesis of the Potassium Quercetin Derivative

A total of 1 mmol of quercetin and 7 mmol of KOH were dissolved in 1 L of methanol. After 3 h of magnetic stirring, the solution was filtered and then concentrated and dried using a vacuum evaporator (pressure 300 mbar, temperature range 40–65 °C). The surface of the substance was ground and designated for testing.

#### 4.2.3. Analysis of Absorption Spectra in the UV-Vis Range

The quercetin derivative was reconstituted in methanol (10 µg·mL^−1^), and then the absorbance in the UV-Vis range (200–800 nm, Metash spectrophotometer) was measured with a measuring increment every 5 nm. The derivative’s absorption spectrum was compared to that of a methanolic quercetin solution.

#### 4.2.4. Analysis of the Antioxidant Activity of Quercetin Derivative

The antioxidant activity of the quercetin derivative was measured using ABTS and DPPH radicals, according to the protocol proposed by Piechowiak et al. [55]. The obtained results were compared with standard solutions of quercetin, ascorbic acid and Trolox and expressed as a percentage of the activity of the mentioned standards.

### 4.3. Measurement of Physiological Parameters

#### 4.3.1. Relative Chlorophyll Content

Measurements were made using a hand-held Chlorophyll Content Meter CCM-200plus (Opti-Sciences, Hudson, NH, USA) calculating an index in CCI units based on absorbance at 650 and 940 nm. These measurements were made on full expanded wheat leaves. 5 leaves per pot were analysed.

#### 4.3.2. Chlorophyll Fluorescence

Measurements of chlorophyll a fluorescence in leaves were performed with an apparatus (Pocket PEA, Hansatech Instruments, King’s Lynn, Norfolk, UK). The maximal available intensity was 3500 μmol which was applied for 1 s with light with a peak wavelength of 627 nm. The first full developed leaves were dark adapted for a period of 30 min using leaf clips which were applied over the adaxial leaf blades [56]. The following parameters were analysed during the study: the maximum quantum yield of primary photochemistry (F_v_/F_0_), the photochemical efficiency of PS II (F_v_/F_m_) and the performance index of PS II (PI). Two measurements of chlorophyll fluorescence were made per pot.

#### 4.3.3. Gas Exchange

A Portable Photosynthesis Measurement System LCpro-SD (ADC BioScientific Ltd., Hoddesdon, UK) was used to determine the gas exchange parameters: net photosynthetic rate (P_N_), transpiration rate (E), stomatal conductance (g_s_) and intercellular CO_2_ concentration (C_i_). When measurements were taken, light intensity was 300 µmol m^−2^ s^−1^ and the leaf chamber temperature was 22 °C. Two leaves were analysed for each pot.

### 4.4. Measurement of Biochemical Parameters

#### 4.4.1. Total Antioxidant Capacity

Total antioxidant capacity was determined by the CUPRAC method using the reduction of copper ions bound in a complex with neocuproine occurring in a neutral environment [57]. Colour intensity changes were measured using the spectrophotometric method. The absorbance of the solutions was measured at the wavelength λ = 450 nm. The analysis was performed three times. The total oxidising capacity was reported as the Trolox equivalent (mg) in 100 g of wheat grain.

#### 4.4.2. Total Content of Polyphenolic Compounds

Total polyphenolic compounds were determined using the Folin-Ciocalteu method [58]. It is based on measuring the absorbance of the complex formed in the molybdenum reduction reaction contained in the Folin-Ciocalteu reagent in an alkaline environment. The absorbance of the solutions was measured at the wavelength λ = 690 nm. The results are expressed as the gallic acid equivalent (mg) contained in 100 g of wheat grains. The measurement was performed in triplicate.

### 4.5. Statistical Analysis

The obtained results were statistically analysed using the Statistica 13.3.0 program (TIBCO Software Inc., Palo Alto, CA, USA). A two-way ANOVA with repeated measurements (time evaluation as a factor) was used. Tukey’s HSD post-hoc test (*p* = 0.05) was used to determine the differences between the mean values of the analysed parameters. The differences between the antioxidant activity against ABTS and DPPH of the potassium quercetin derivative, expressed as a percentage of the activity of the standard substance, were performed by the t-Student’s test (*p* = 0.05).

## 5. Conclusions

The conducted experiment showed the stimulating effect of quercetin derivative solutions on the course of physiological and biochemical processes taking place in wheat seedlings. It has been shown that with an increase in quercetin derivative concentrations, the value of most of the analysed parameters increases compared to the control. As a result of the experiment, a positive effect of quercetin derivative solutions on the analysed physiological parameters at subsequent measurement dates was also found. In most cases, no significant differences in the values of the parameters were observed between the applied concentrations of 0.5% and 1.0% as well as 3.0% and 5.0%.

Preliminary studies allowed for the selection of the most favourable dose (3.0%) that could be used to test the effect of the external application of this flavonoid on wheat plants exposed to various biotic and abiotic stresses. Such knowledge may contribute to the introduction of sustainable agricultural practices in the future.

## Figures and Tables

**Figure 1 ijms-22-06882-f001:**
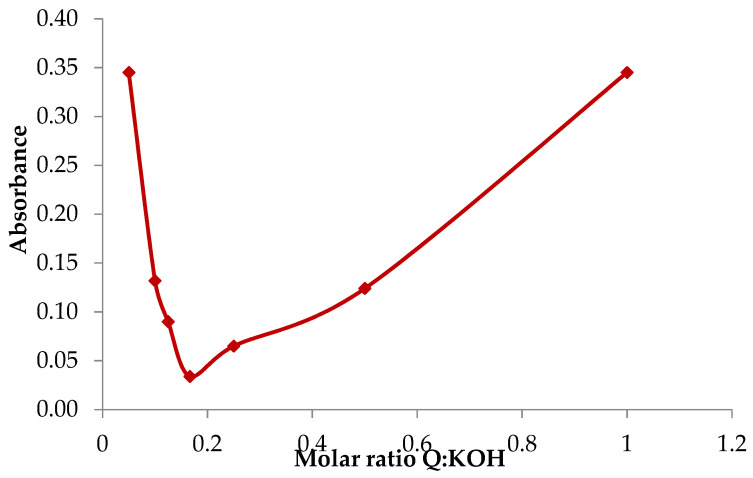
Determination of optimal molar ratios between the reagents by Job’s method.

**Figure 2 ijms-22-06882-f002:**
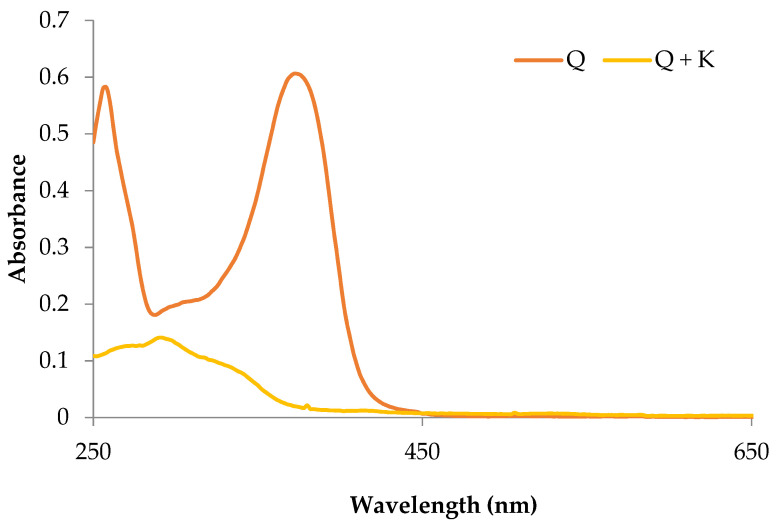
UV-vis spectrum of potassium quercetin derivative and quercetin. Q—quercetin methanol solution with a concentration of 0.1 mg/mL. Q + K—methanolic solution of potassium quercetin derivative with a concentration of 0.1 mg/mL.

**Figure 3 ijms-22-06882-f003:**
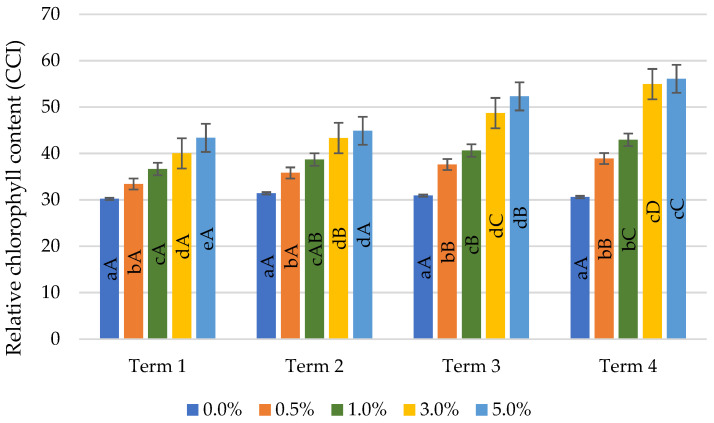
Effect of quercetin derivative concentrations and terms of measurement on relative chlorophyll content (Term 1—first day after the first treatment, Term 2—seventh days after the first treatment, Term 3—first day after the second treatment, Term 4—seventh days after the second treatment). Capital letters indicate significant differences between the means at measurement terms for each quercetin derivative concentrations, lowercase letters indicate significant differences between the means at respective measurement terms according to ANOVA (followed by Tukey’s HSD test, *p* = 0.05).

**Figure 4 ijms-22-06882-f004:**
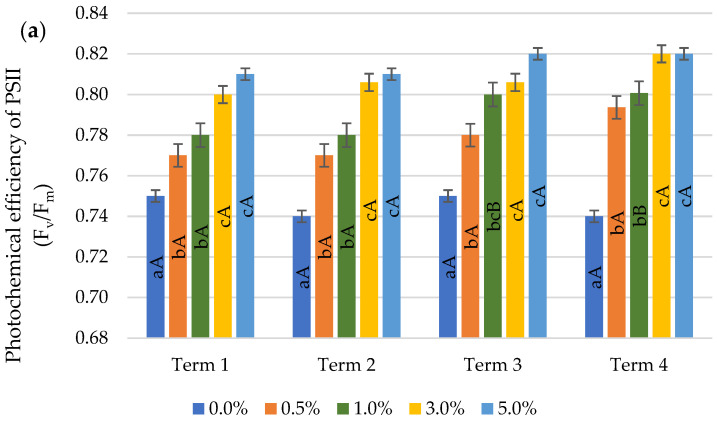
Effect of quercetin derivative concentrations and terms of measurement on chlorophyll fluorescence parameters: (**a**) photochemical efficiency of PS II (F_v_/F_m_); (**b**) maximum quantum yield of primary photochemistry (F_v_/F_0_) and (**c**) performance index of PS II (PI). (Term 1—first day after the first treatment, Term 2—seventh days after the first treatment, Term 3—first day after the second treatment, Term 4—seventh days after the second treatment). Capital letters indicate significant differences between the means at measurement terms for each quercetin derivative concentrations, lowercase letters indicate significant differences between the means at respective measurement terms according to ANOVA (followed by Tukey’s HSD test, *p* = 0.05).

**Figure 5 ijms-22-06882-f005:**
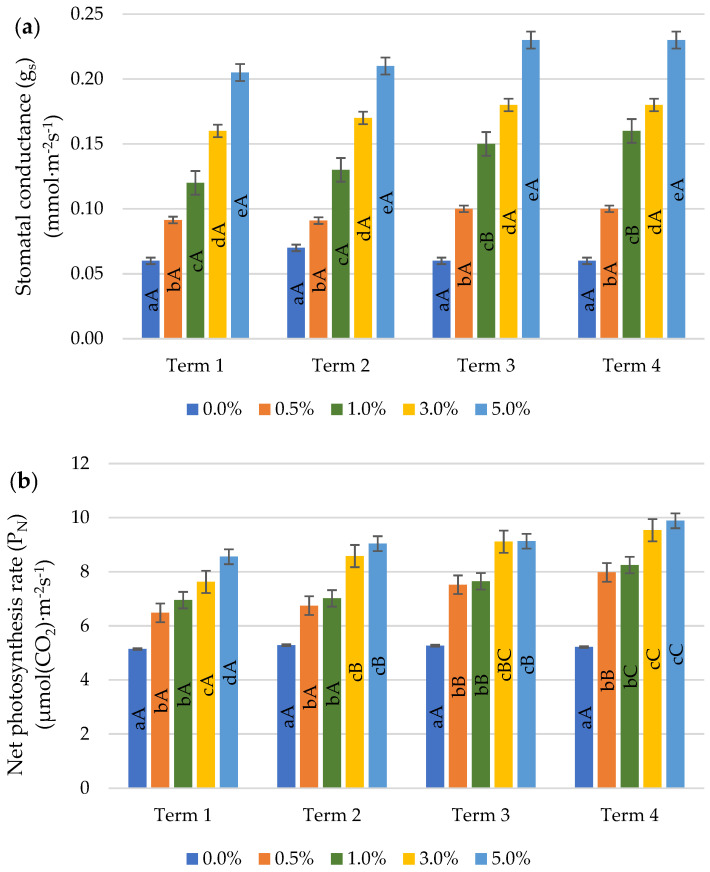
Effect of quercetin derivative concentrations and terms of measurement on gas exchange parameters: (**a**) stomatal conductance (g_s_); (**b**) net photosynthesis rate (P_N_); (**c**) transpiration rate (E) and (**d**) intercellular CO_2_ concentration (C_i_). (Term 1—first day after the first treatment, Term 2—seventh days after the first treatment, Term 3—first day after the second treatment, Term 4—seventh days after the second treatment). Capital letters indicate significant differences between the means at measurement terms for each quercetin derivative concentrations, lowercase letters indicate significant differences between the means at respective measurement terms according to ANOVA (followed by Tukey’s HSD test, *p* = 0.05).

**Figure 6 ijms-22-06882-f006:**
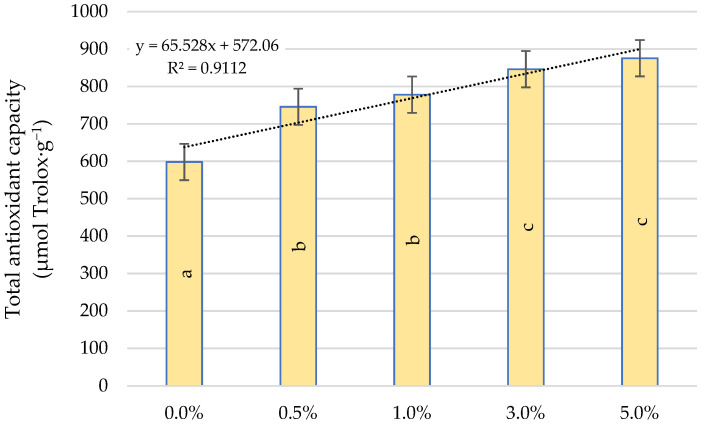
Effect of quercetin derivative concentrations on total antioxidant capacity. Different letters indicate significant differences between each quercetin derivative concentrations, according to ANOVA (followed by Tukey’s HSD test, *p* = 0.05).

**Figure 7 ijms-22-06882-f007:**
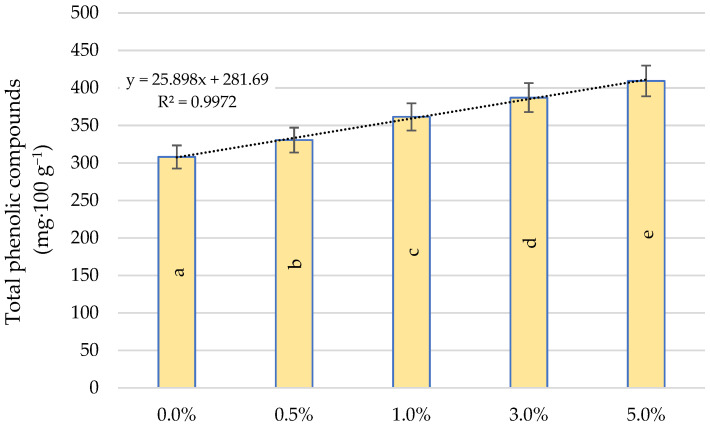
Effect of quercetin derivative concentrations on total phenolic compounds. Different letters indicate significant differences between each quercetin derivative concentrations, according to ANOVA (followed by Tukey’s HSD test, *p* = 0.05).

**Table 1 ijms-22-06882-t001:** Antioxidant activity against ABTS and DPPH of the potassium quercetin derivative, expressed as a percentage of the standard substance activity. Lowercase letters indicate significant differences between the means at respective measurement according to t-Student’s test (*p* = 0.05).

Method	Antioxidant Activity of the Preparation Expressed as:		
	% of Quercetin Activity	% of Trolox Activity	% of Ascorbic Acid Activity
ABTS	67.02 ± 4.21 a	104.01 ± 1.75 a	145.34 ± 4.56 b
DPPH	33.43 ± 2.26 a	94.12 ± 6.43 a	110.23 ± 4.85 b

## Data Availability

The data presented in this study are available on request from the corresponding author.

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
