# Peer review of "The Effect of Exogenous Application of Quercetin Derivative Solutions on the Course of Physiological and Biochemical Processes in Wheat Seedlings"

_ijms, 2021, doi:10.3390/ijms22136882_

Round 1

Reviewer 1 Report

The authors did not use appropriate methods to quantify and identify the phenolics. Papers based primarily on total phenolic assays are not acceptable for a high impact factor journal such as IJMS. At this stage, this is preliminary study. Therefore, at this point this manuscript does not have merit for publication. 

Author Response

Response to Reviewer 1 Comments

Thank you for the constructive and informative comments of the Reviewer. Below is our response to the concerns raised during review process.

Point: The authors did not use appropriate methods to quantify and identify the phenolics. Papers based primarily on total phenolic assays are not acceptable for a high impact factor journal such as IJMS. At this stage, this is preliminary study. Therefore, at this point this manuscript does not have merit for publication. 

Response: Thank you for your comment. The major aim of this study was to determine the effect of potassium quercetin derivative spraying on the physiological and selected biochemical parameters of wheat seedlings and to select the optimal dose. In the studies used only derivatives of one phenolic factor - a derivative of quercetin as a stimulating factor. In the authors' opinion, determining the profile of phenolic compounds is not necessary because the plants have been enriched only in quercetin, which directly affect the total content of this group of compounds. It has been shown that the total polyphenol content linearly dependent on the dose of quercetin. In the near future, this research will be continued and the effect of quercetin derivative on the oxidative stress markers (ROS generation, antioxidant enzymes activity, mitochondria activity) and phenolic compounds profile will be investigated.

Reviewer 2 Report

Jańczak-Pieniążek et al. prepared a manuscript “The Effect of Exogenous Application of Quercetin Derivative Solutions on the Course of Physiological and Biochemical Processes in Wheat Seedlings”. Study is well designed, and results obtained are promising for the concept of quercetin use as a biopesticide/plant regulator. My major concern is control. Quercetin per se can interfere with many of the analysis conducted e.g. antioxidant properties. Control 0% of quercetin is not adequate. It would be prudent to use sample + standard addition of quercetin to assess if quercetin indeed induces any changes within the plant system or the result is only consequence of exogenous quercetin addition.

Author Response

Response to Reviewer 2 Comments

Point: Jańczak-Pieniążek et al. prepared a manuscript “The Effect of Exogenous Application of Quercetin Derivative Solutions on the Course of Physiological and Biochemical Processes in Wheat Seedlings”. Study is well designed, and results obtained are promising for the concept of quercetin use as a biopesticide/plant regulator. My major concern is control. Quercetin per se can interfere with many of the analysis conducted e.g. antioxidant properties. Control 0% of quercetin is not adequate. It would be prudent to use sample + standard addition of quercetin to assess if quercetin indeed induces any changes within the plant system or the result is only consequence of exogenous quercetin addition.

Response: Thank you for the constructive and informative comments of the Reviewer. We did not use pure quercetin, since its application is difficult because the compound is insoluble in water. This is why we used potassium quercetin, because this derivative is more soluble in water.

Reviewer 3 Report

The article by Marta Jańczak-Pieniążek and co-authors "The Effect of Exogenous Application of Quercetin Derivative Solutions on the Course of Physiological and Biochemical Processes in Wheat Seedlings" is devoted to the assessment of physiology parameters of wheat plants treated by the quercetin derivative. The work was done at a high methodological level, however, it requires adjustments. This study will be of interest to a narrow circle of researchers involved in phytochemistry and redox biology study.

  1. Introduction

- Line 31-34 Common wheat (Triticum aestivum L.) is the most widely cultivated species in the world, representing the main source of calories and protein for the world's population. Therefore, at present, food security largely depends on the increased production of cereals, in particular wheat [1].

Despite the fact that the authors refer to a highly impactful publication and, in particular, to a highly cited article, I believe that over the past 8 years there has been a sufficient number of studies confirming this thesis.

In addition, a link to the reports of the Food and Agriculture Organization of the United Nations over the past few years would be appropriate here.

- Line 34- 37 During cultivation, wheat is exposed to both biotic and abiotic stress factors, which damage plants leading to inhibition of their growth and reduction of grain yield [2]. As a result of oxidative stress, cell organelles and cell membranes are damaged, which can eventually lead to cell death.

There is a lack of logical transition between these two sentences. Although the link between abiotic and biotic stressors and oxidative stress seems clear, this should be pointed out.

- Line 54-56 This compound is commonly found in various vegetables, seeds, nuts, teas and red wine and has a positive effect on human health, providing protection against many diseases and aging of the body.

Not relevant to current research.

- Line 60- 62 Quercetin plays an important role in the process of protecting plants against the effects of stress such as: UV radiation or osmotic stress, as has been confirmed in numerous studies [16-18].

I suppose that osmoregulation will be more characteristic of quercetin glycosides. Glycosylation is most often carried out in the cytoplasm, and the deposition of these substances in vacuoles. Thus, these compounds are spatially and functionally separated from chloroplasts.

- Line 78 -80 Therefore, more and more emphasis is being placed on the use of safe agents - biopesticides, which could contain flavonoids - including quercetin [24].

The described effect of flavonoids, and in particular of quercitin (influence on auxin signaling), indicates that this group of compounds play the role of a biostimulant, but not a pesticide.

- Line 80-81 However, information on the effect of the external application of quercetin on the physiological processes occurring in wheat seedlings is insufficient [25].

I suppose that the link used here is erroneous, the information in the article does not correspond to this thesis.

Despite the fact that this assumption looks quite reasonable (though not supported by the corresponding link), a cursory search showed a fairly large number of works indicating the effect of exogenous phenolic substances on the metabolic processes of plants.

Novak, K., Chovanec, P., Škrdleta, V., Kropáčová, M., Lisá, L., & Němcová, M. (2002). Effect of exogenous flavonoids on nodulation of pea (Pisum sativum L.). Journal of Experimental Botany, 53 (375), 1735-1745.

Lu, X. F., Zhang, H., Lyu, S. S., Du, G. D., Wang, X. Q., Wu, C. H., & Lyu, D. G. (2018). Effects of exogenous phenolic acids on photosystem functions and photosynthetic electron transport rate in strawberry leaves. Photosynthetica, 56 (2), 616-622.

Wang, Z., Ma, L., Zhang, X., Xu, L., Cao, J., & Jiang, W. (2015). The effect of exogenous salicylic acid on antioxidant activity, bioactive compounds and antioxidant system in apricot fruit. Scientia Horticulturae, 181, 113-120.

There is also a pool of works demonstrating the effect of exogenous precursors of flavonoids (for example, aromatic amino acids) on the synthetic processes of plants.

Aghdam, M. S., Moradi, M., Razavi, F., & Rabiei, V. (2019). Exogenous phenylalanine application promotes chilling tolerance in tomato fruits during cold storage by ensuring supply of NADPH for activation of ROS scavenging systems. Scientia Horticulturae, 246, 818-825.

Feduraev, P., Skrypnik, L., Riabova, A., Pungin, A., Tokupova, E., Maslennikov, P., & Chupakhina, G. (2020). Phenylalanine and tyrosine as exogenous precursors of wheat (triticum aestivum L.) secondary metabolism through PAL-associated pathways. Plants, 9 (4), 476.

Md-Mustafa, N. D., Khalid, N., Gao, H., Peng, Z., Alimin, M. F., Bujang, N., ... & Othman, R. Y. (2014). Transcriptome profiling shows gene regulation patterns in a flavonoid pathway in response to exogenous phenylalanine in Boesenbergia rotunda cell culture. BMC genomics, 15 (1), 1-25.

Perhaps a number of materials from these works will help the authors in discussing of the results.

- Line 84-85 The obtained results will allow us to assess the usefulness of this flavonoid as a preventive agent protecting plants against biotic and abiotic stresses.

The work did not assess the influence of factors on experimental plants.

- The introduction describes the effect of quercetin as a regulator of auxin transport, however, this effect is not further recalled, not when describing the results, not when discussing them.

  2. Results:

- In the figures, an error bar is introduced, but it is not indicated which error was estimated by the authors.

- Picture 1

Remove the frame around the chart (this also applies to other figures)

The columns show absolutized values (probably averages), however, error bars are also set, this format of data designation is misleading

On the ordinate axis, the values are written in tenths, but they are not significant

- 2.5. Total Antioxidant Capacity

The data presented in this section is practically not described.

- Line 252 "As a result of its use, the stimulating effect of this flavonoid"

It is more appropriate to speak, after all, about a derivative of this flavonoid.

Why were the data in this section recalculated for trolox, and not for ascorbic acid (previous tests showed a greater antioxidant activity in terms of this compound)?

 3. Discussion

- The discussion focuses on the physiological effects of quercitin. However, the work investigates the  effect of its inorganic derivative. Thus, it is not entirely clear how comparable such data are?

- Line 301-312

This paragraph describes the results obtained without attempting to rethink them.

- Line 366-369 The biosynthesis of phenolic compounds is related to the activation of key biosynthetic enzymes and the upregulation of the key genes of the phenylpropanoid branch - including the phenylalanine ammonia-lyase enzyme (PAL, EC 4.3.1.5), which is involved in the biosynthesis of antioxidant phenolic compounds [45].

PAL regulates the synthesis of phenolic compounds upstream of the metabolic pathway. Probably, an increase in the exogenous pool of quercetin will lead to a decrease in its activity (by the reciprocal mechanism), or the conversion products will be redistributed to the formation of lignin. However, the final paragraph of the discussion raises an interesting question of how exogenous quercetin can affect the balance of the entire phenolic complex.

  4. Methods

- What is the reason for the choice of such a range of concentrations of quercetin derivatives?

For a very large pool of data, a concentration of 5% quercetin derivative is stimulating. Thus, it becomes necessary to expand the range of concentrations in order to make sure whether a given concentration (5%) is "peak", and all subsequent ones will show a similar effect (reaching a plateau), suppress physiological processes or stimulate even more effectively.

- Line 386-387 Two spraying treatments were performed: 10 and 17 days after the 4- leaf stage with a hand sprayer.

I think it is better to indicate the phase of growth in accordance with international scales (BBCH-scale, Feekes scale).

- What is the reason for the periodization of the treatment of plants with solutions of quercitin?

- Line 443-444 When measurements were taken, light intensity was 1500 mol m-2 s-1 and the leaf chamber temperature was 26 ° C.

Why are the conditions for this analysis different from the basic growing conditions?

Would recommend that authors express light intensity in the same units.

- Line 446 -448 4.4. Measurement of biochemical parameters

4.4.1. Total antioxidant capacity

What caused the choice of the method? why the antioxidant properties of the extract were not evaluated by ABTS and DPPH methods, with the help of which the antioxidant properties of the obtained potassium quercetin?

Author Response

Response to Reviewer 3 Comments

Thank you for the constructive and informative comments of the Reviewer. Below are our point-by-point responses to the concerns raised during review process.

Point 1: Line 31-34 Common wheat (Triticum aestivum L.) is the most widely cultivated species in the world, representing the main source of calories and protein for the world's population. Therefore, at present, food security largely depends on the increased production of cereals, in particular wheat [1].

Despite the fact that the authors refer to a highly impactful publication and, in particular, to a highly cited article, I believe that over the past 8 years there has been a sufficient number of studies confirming this thesis.

In addition, a link to the reports of the Food and Agriculture Organization of the United Nations over the past few years would be appropriate here.

Response 1:

According to Reviewer’s suggestion we have added citations from recent years and information from the Food and Agriculture Organization of the United Nations regarding the current area of wheat cultivation in the world L: 32-36.

  •  http://www.fao.org/faostat/en/#data/QC (accessed on 8 June 2021)
  • Tadesse, W.; Sanchez-Garcia, M.; Assefa, S.G.; Amri, A.; Bishaw Z.; Ogbonnaya, F.C.; Baum, M. Genetic gains in wheat breeding and its role in feeding the world. Crop Breed Genet Genom. 2019, 1, 190005. doi: 10.20900/cbgg20190005
  • Hyles, J.; Bloomfield, M.T.; Hunt, J.R.; Trethowan, R.M.; Trevaskis, B. Phenology and related traits for wheat adaptation. Heredity. 2020, 125, 417–430 doi: 10.1038/s41437-020-0320-1

Point 2: Line 34-37 During cultivation, wheat is exposed to both biotic and abiotic stress factors, which damage plants leading to inhibition of their growth and reduction of grain yield [2]. As a result of oxidative stress, cell organelles and cell membranes are damaged, which can eventually lead to cell death.

There is a lack of logical transition between these two sentences. Although the link between abiotic and biotic stressors and oxidative stress seems clear, this should be pointed out.

Response 2: According to Reviewer’s suggestion we have changed these sentences. L: 36-42

Point 3: Line 54-56 This compound is commonly found in various vegetables, seeds, nuts, teas and red wine and has a positive effect on human health, providing protection against many diseases and aging of the body.

Not relevant to current research.

Response 3: We removed this sentence according to Reviewer’s suggestion.

Point 4: Line 60- 62 Quercetin plays an important role in the process of protecting plants against the effects of stress such as: UV radiation or osmotic stress, as has been confirmed in numerous studies [16-18].

I suppose that osmoregulation will be more characteristic of quercetin glycosides. Glycosylation is most often carried out in the cytoplasm, and the deposition of these substances in vacuoles. Thus, these compounds are spatially and functionally separated from chloroplasts.

Response 4: We have added this information in the text of manuscript. L: 69-73

Point 5: Line 78 -80 Therefore, more and more emphasis is being placed on the use of safe agents - biopesticides, which could contain flavonoids - including quercetin [24].

The described effect of flavonoids, and in particular of quercitin (influence on auxin signaling), indicates that this group of compounds play the role of a biostimulant, but not a pesticide.

Response 5: We changed “biopesticides” to “biostimulants” according to Reviewer’s suggestion. L: 91.

Point 6: Line 80-81 However, information on the effect of the external application of quercetin on the physiological processes occurring in wheat seedlings is insufficient [25].

I suppose that the link used here is erroneous, the information in the article does not correspond to this thesis.

Despite the fact that this assumption looks quite reasonable (though not supported by the corresponding link), a cursory search showed a fairly large number of works indicating the effect of exogenous phenolic substances on the metabolic processes of plants.

Novak, K., Chovanec, P., Škrdleta, V., Kropáčová, M., Lisá, L., & Němcová, M. (2002). Effect of exogenous flavonoids on nodulation of pea (Pisum sativum L.). Journal of Experimental Botany, 53 (375), 1735-1745.

Lu, X. F., Zhang, H., Lyu, S. S., Du, G. D., Wang, X. Q., Wu, C. H., & Lyu, D. G. (2018). Effects of exogenous phenolic acids on photosystem functions and photosynthetic electron transport rate in strawberry leaves. Photosynthetica, 56 (2), 616-622.

Wang, Z., Ma, L., Zhang, X., Xu, L., Cao, J., & Jiang, W. (2015). The effect of exogenous salicylic acid on antioxidant activity, bioactive compounds and antioxidant system in apricot fruit. Scientia Horticulturae, 181, 113-120.

There is also a pool of works demonstrating the effect of exogenous precursors of flavonoids (for example, aromatic amino acids) on the synthetic processes of plants.

Aghdam, M. S., Moradi, M., Razavi, F., & Rabiei, V. (2019). Exogenous phenylalanine application promotes chilling tolerance in tomato fruits during cold storage by ensuring supply of NADPH for activation of ROS scavenging systems. Scientia Horticulturae, 246, 818-825.

Feduraev, P., Skrypnik, L., Riabova, A., Pungin, A., Tokupova, E., Maslennikov, P., & Chupakhina, G. (2020). Phenylalanine and tyrosine as exogenous precursors of wheat (triticum aestivum L.) secondary metabolism through PAL-associated pathways. Plants, 9 (4), 476.

Md-Mustafa, N. D., Khalid, N., Gao, H., Peng, Z., Alimin, M. F., Bujang, N., ... & Othman, R. Y. (2014). Transcriptome profiling shows gene regulation patterns in a flavonoid pathway in response to exogenous phenylalanine in Boesenbergia rotunda cell culture. BMC genomics, 15 (1), 1-25.

Perhaps a number of materials from these works will help the authors in discussing of the results.

Response 6: Thank you very much for this valuable attention and presentation of many literature items. We used these references for quoting. In the manuscript is written, that there are few papers describing the effect of external application of phenolic substances on the physiological processes occurring in wheat seedlings. I agree with the Reviewer, that is a large literature on the subject, but the manuscript was written specifically that it is a wheat seedlings. We have rewritten this sentence L: 92-96.

Point 7: Line 84-85 The obtained results will allow us to assess the usefulness of this flavonoid as a preventive agent protecting plants against biotic and abiotic stresses.

The work did not assess the influence of factors on experimental plants.

Response 7: We added information that, the obtained results will allow in future studies to assess the usefulness of this flavonoid as a preventive measure protecting plants against biotic and abiotic stresses L: 99.

Point 8: The introduction describes the effect of quercetin as a regulator of auxin transport, however, this effect is not further recalled, not when describing the results, not when discussing them.

Response 8: According to Reviewer’s suggestion we removed information about the effect of quercetin as a regulator of auxin transport.

Point 9: 2. Results:

- In the figures, an error bar is introduced, but it is not indicated which error was estimated by the authors.

Response 9: The results are presented as mean values with standard deviation for the given sample. However, we guess that reporting the results in this way can be misleading, so the error bars in the Figures 3, 4, 5,6, 7 have been removed.

Point 10: Picture 1

Remove the frame around the chart (this also applies to other figures)

The columns show absolutized values (probably averages), however, error bars are also set, this format of data designation is misleading

On the ordinate axis, the values are written in tenths, but they are not significant

Response 10: The drawing borders have been removed (in all Figures). The format of the values on the ordinate axis has also been changed.

Point 11:  2.5. Total Antioxidant Capacity

The data presented in this section is practically not described.

Response 11: According to Reviewer’s suggestion we extended the description of the results presented in (2.5. Total Antioxidant Capacity section and 2.5. Total Phenolic Compounds) L: 267-270 and 291-293.

Point 12: Line 252 "As a result of its use, the stimulating effect of this flavonoid"

It is more appropriate to speak, after all, about a derivative of this flavonoid.

Why were the data in this section recalculated for trolox, and not for ascorbic acid (previous tests showed a greater antioxidant activity in terms of this compound)?

Response 12: Antioxioxidant activity (AA) of the samples was expressed as trolox equivalent because AA of quercetin derivative was comparable to trolox – see: Table 1.

Point 13: 3. Discussion

- The discussion focuses on the physiological effects of quercitin. However, the work investigates the  effect of its inorganic derivative. Thus, it is not entirely clear how comparable such data are?

Response 13: Quercetin has low water solubility, hence the need to synthesize a derivative. The potassium derivative was chosen because K it is an important fertilizing element. However, after spraying the plant with quercetin derivative, it will directly release quercetin due to the presence of organic acids on the plant surface that take over the potassium ions.

Point 14: - Line 301-312

This paragraph describes the results obtained without attempting to rethink them.

Response 14: On lines 320-325 there is an introduction to the discussion synthetically describing the most important results. The rest of the Discussion section describes the rethink of the results obtained.

Point 15: - Line 366-369 The biosynthesis of phenolic compounds is related to the activation of key biosynthetic enzymes and the upregulation of the key genes of the phenylpropanoid branch - including the phenylalanine ammonia-lyase enzyme (PAL, EC 4.3.1.5), which is involved in the biosynthesis of antioxidant phenolic compounds [45].

PAL regulates the synthesis of phenolic compounds upstream of the metabolic pathway. Probably, an increase in the exogenous pool of quercetin will lead to a decrease in its activity (by the reciprocal mechanism), or the conversion products will be redistributed to the formation of lignin. However, the final paragraph of the discussion raises an interesting question of how exogenous quercetin can affect the balance of the entire phenolic complex.

Response 15: Application of a potassium quercetin derivative can reduce the level of ROS in the plant cell, which can reduce the expression of enzymes dependent on the level of oxidative stress, e.g. phenylalanine ammonia-lyase and polyphenol oxidase (PPO) This may lead to inhibition of the biosynthesis of new polyphenol compounds, but also to inhibition of their degradation by PPO. However, in order to fully understand these mechanisms, it is necessary to conduct deeper analysis at a molecular level.  

Point 16: 4. Methods

- What is the reason for the choice of such a range of concentrations of quercetin derivatives?

For a very large pool of data, a concentration of 5% quercetin derivative is stimulating. Thus, it becomes necessary to expand the range of concentrations in order to make sure whether a given concentration (5%) is "peak", and all subsequent ones will show a similar effect (reaching a plateau), suppress physiological processes or stimulate even more effectively.

Response 16: In this research, we used the concentration range from 0.5-5% of the quercetin derivative, because in solutions above 5% we observed the precipitation of the substance from the solution.

Point 17: Line 386-387 Two spraying treatments were performed: 10 and 17 days after the 4- leaf stage with a hand sprayer.

I think it is better to indicate the phase of growth in accordance with international scales (BBCH-scale, Feekes scale).

Response 17: We indicated this phase in accordance with BBCH-scale. L: 406-407

Point 18:  What is the reason for the periodization of the treatment of plants with solutions of quercitin?

Response 18: We performed physiological measurements on wheat seedlings treated with quercetin derivative solutions in 4 terms. The reason for this was to study the influence of the quercetin derivative application on the course of physiological and biochemical processes 1 day and 7 days after the spraying. On the other hand, repeating the spraying treatment was to check the effectiveness of this flavonoid and whether the repeated treatment would have a toxic or stimulating effect on wheat seedlings.

Point 19: Line 443-444 When measurements were taken, light intensity was 1500 mol m-2 s-1 and the leaf chamber temperature was 26° C.

Why are the conditions for this analysis different from the basic growing conditions?

Would recommend that authors express light intensity in the same units.

Response 19: In the methodology, the values of light intensity and leaf chamber temperature during the measurement of gas exchange were given mistakenly. In accordance with the Reviewer's remark, the units were also made uniform. L: 400 and 464 - 465

Point 20:  Line 446 -448 4.4. Measurement of biochemical parameters

4.4.1. Total antioxidant capacity

What caused the choice of the method? why the antioxidant properties of the extract were not evaluated by ABTS and DPPH methods, with the help of which the antioxidant properties of the obtained potassium quercetin?

Response 20: We made an attempt to assess the antioxidant activity of wheat seedlings with ABTS and DPPH radicals, but the green colour of the extracts derived from chlorophyll made the measurement difficult (increased the absorption). Therefore, the antioxidant activity of the extracts was measured using the CUPRAC method.

Round 2

Reviewer 1 Report

I understand that the authors will address the phenolic profile as a result of potassium quercetin solution application. However, they also did not identify the derivative of quercetin formed when mixed with KOH.

Considering Figure 2, although this remains to be confirmed in the future, based on the literature (Fuentes et al. http://dx.doi.org/10.1021/acs.jafc.7b05214), the authors could suggest that benzofuranone may be the active molecule.

The authors could also use this opportunity to mention that this quercetin metabolite has shown a high potency against oxidative stress in cell model systems (Fuentes et al., https://doi.org/10.1021/acs.jafc.0c07085). 

Author Response

Response to Reviewer 1 Comments

Thank you for the constructive and informative comments of the Reviewer. Below is our response to the concerns raised during review process.

Point 1: I understand that the authors will address the phenolic profile as a result of potassium quercetin solution application. However, they also did not identify the derivative of quercetin formed when mixed with KOH.

Response 2: Applying the reaction of KOH with quercetin only causes the loss of phenolic protons. This process was investigated by measuring the UV spectra, which led to the determination of the stoichiometry of this reaction. No other changes in the structure are expected, so we think that no other spectra analysis of the product is needed.

Point 2: Considering Figure 2, although this remains to be confirmed in the future, based on the literature (Fuentes et al. http://dx.doi.org/10.1021/acs.jafc.7b05214), the authors could suggest that benzofuranone may be the active molecule. The authors could also use this opportunity to mention that this quercetin metabolite has shown a high potency against oxidative stress in cell model systems (Fuentes et al., https://doi.org/10.1021/acs.jafc.0c07085).

Response 2: According to Reviewer’s suggestion we added information about benzofuranone. L: 318-324.

Reviewer 3 Report

  1. «- In the figures, an error bar is introduced, but it is not indicated which error was estimated by the authors.

Response 9: The results are presented as mean values with standard deviation for the given sample. However, we guess that reporting the results in this way can be misleading, so the error bars in the Figures 3, 4, 5,6, 7 have been removed.»

I would suggest doing it differently. Remove average values from histogram bars and reapply error bars. And the mean values with standard deviations should be used in the text when describing the results.

  1. «Why were the data in this section recalculated for trolox, and not for ascorbic acid (previous tests showed a greater antioxidant activity in terms of this compound)?

Response 12: Antioxioxidant activity (AA) of the samples was expressed as trolox equivalent because AA of quercetin derivative was comparable to trolox – see: Table 1.»

I guess it is comparable to what you compare it to. Quercitin is a water-soluble compound and it would be more correct to recalculate its activity for water-soluble ascorbic acid

  1. «Point 15:- Line 366-369 The biosynthesis of phenolic compounds is related to the activation of key biosynthetic enzymes and the upregulation of the key genes of the phenylpropanoid branch - including the phenylalanine ammonia-lyase enzyme (PAL, EC 4.3.1.5), which is involved in the biosynthesis of antioxidant phenolic compounds [45].

PAL regulates the synthesis of phenolic compounds upstream of the metabolic pathway. Probably, an increase in the exogenous pool of quercetin will lead to a decrease in its activity (by the reciprocal mechanism), or the conversion products will be redistributed to the formation of lignin. However, the final paragraph of the discussion raises an interesting question of how exogenous quercetin can affect the balance of the entire phenolic complex.

Response 15: Application of a potassium quercetin derivative can reduce the level of ROS in the plant cell, which can reduce the expression of enzymes dependent on the level of oxidative stress, e.g. phenylalanine ammonia-lyase and polyphenol oxidase (PPO) This may lead to inhibition of the biosynthesis of new polyphenol compounds, but also to inhibition of their degradation by PPO. However, in order to fully understand these mechanisms, it is necessary to conduct deeper analysis at a molecular level.»

This is a fairly fundamental point, and in the discussion you can suggest which molecular mechanisms may be involved in the induction of PAL (this will be in the trend of the magazine). I suggest that the authors refer back to the links I recommended.

There is also a pool of works demonstrating the effect of exogenous precursors of flavonoids (for example, aromatic amino acids) on the synthetic processes of plants.

Aghdam, M. S., Moradi, M., Razavi, F., & Rabiei, V. (2019). Exogenous phenylalanine application promotes chilling tolerance in tomato fruits during cold storage by ensuring supply of NADPH for activation of ROS scavenging systems. Scientia Horticulturae, 246, 818-825.

Feduraev, P., Skrypnik, L., Riabova, A., Pungin, A., Tokupova, E., Maslennikov, P., & Chupakhina, G. (2020). Phenylalanine and tyrosine as exogenous precursors of wheat (triticum aestivum L.) secondary metabolism through PAL-associated pathways. Plants, 9 (4), 476.

Md-Mustafa, N. D., Khalid, N., Gao, H., Peng, Z., Alimin, M. F., Bujang, N., ... & Othman, R. Y. (2014). Transcriptome profiling shows gene regulation patterns in a flavonoid pathway in response to exogenous phenylalanine in Boesenbergia rotunda cell culture. BMC genomics, 15 (1), 1-25.

Author Response

Response to Reviewer 3 Comments

Thank you for the constructive and informative comments of the Reviewer. Below are our point-by-point responses to the concerns raised during review process.

Point 1: «- In the figures, an error bar is introduced, but it is not indicated which error was estimated by the authors.

Response 9: The results are presented as mean values with standard deviation for the given sample. However, we guess that reporting the results in this way can be misleading, so the error bars in the Figures 3, 4, 5,6, 7 have been removed.»

I would suggest doing it differently. Remove average values from histogram bars and reapply error bars. And the mean values with standard deviations should be used in the text when describing the results.

Response 1: According to the Reviewer's comments, the figures have been improved.

Point 2: «Why were the data in this section recalculated for trolox, and not for ascorbic acid (previous tests showed a greater antioxidant activity in terms of this compound)?

Response 12: Antioxioxidant activity (AA) of the samples was expressed as trolox equivalent because AA of quercetin derivative was comparable to trolox – see: Table 1.»

I guess it is comparable to what you compare it to. Quercitin is a water-soluble compound and it would be more correct to recalculate its activity for water-soluble ascorbic acid

Response 2: In order to prove that the antioxidant activity of the quercetin derivative does not differ significantly from the trolox, we performed an additional statistical analysis (Student's t-test) - see table 1 and subsection statistical analysis. Besides, trolox is a widely used standard in the analysis of an antioxidant properties, in relation to both polar and non-polar antioxidants. At the moment, we cannot express the antioxidant activity as an equivalent of ascorbic acid, because it is necessary to make a calibration curve and repeat the analysis. Moreover, the use of ascorbic acid as a standard will not change the relationships that occurred between the samples, but only the values of the antioxidant activity. However, in the future we will pay attention to the issue raised by the Reviewer.

Point 3: «Point 15:- Line 366-369 The biosynthesis of phenolic compounds is related to the activation of key biosynthetic enzymes and the upregulation of the key genes of the phenylpropanoid branch - including the phenylalanine ammonia-lyase enzyme (PAL, EC 4.3.1.5), which is involved in the biosynthesis of antioxidant phenolic compounds [45].

PAL regulates the synthesis of phenolic compounds upstream of the metabolic pathway. Probably, an increase in the exogenous pool of quercetin will lead to a decrease in its activity (by the reciprocal mechanism), or the conversion products will be redistributed to the formation of lignin. However, the final paragraph of the discussion raises an interesting question of how exogenous quercetin can affect the balance of the entire phenolic complex.

Response 15: Application of a potassium quercetin derivative can reduce the level of ROS in the plant cell, which can reduce the expression of enzymes dependent on the level of oxidative stress, e.g. phenylalanine ammonia-lyase and polyphenol oxidase (PPO) This may lead to inhibition of the biosynthesis of new polyphenol compounds, but also to inhibition of their degradation by PPO. However, in order to fully understand these mechanisms, it is necessary to conduct deeper analysis at a molecular level.»

This is a fairly fundamental point, and in the discussion you can suggest which molecular mechanisms may be involved in the induction of PAL (this will be in the trend of the magazine). I suggest that the authors refer back to the links I recommended.

There is also a pool of works demonstrating the effect of exogenous precursors of flavonoids (for example, aromatic amino acids) on the synthetic processes of plants.

Aghdam, M. S., Moradi, M., Razavi, F., & Rabiei, V. (2019). Exogenous phenylalanine application promotes chilling tolerance in tomato fruits during cold storage by ensuring supply of NADPH for activation of ROS scavenging systems. Scientia Horticulturae, 246, 818-825.

Feduraev, P., Skrypnik, L., Riabova, A., Pungin, A., Tokupova, E., Maslennikov, P., & Chupakhina, G. (2020). Phenylalanine and tyrosine as exogenous precursors of wheat (triticum aestivum L.) secondary metabolism through PAL-associated pathways. Plants, 9 (4), 476.

Md-Mustafa, N. D., Khalid, N., Gao, H., Peng, Z., Alimin, M. F., Bujang, N., ... & Othman, R. Y. (2014). Transcriptome profiling shows gene regulation patterns in a flavonoid pathway in response to exogenous phenylalanine in Boesenbergia rotunda cell culture. BMC genomics, 15 (1), 1-25.

Response 3:

The authors thank the Reviewer for the wide spectrum of proposed publications. Most of them were used to enrich the content of the discussion. According to Reviewers suggestion in lines: 395-407 we added information about molecular mechanisms involved in the induction of PAL.

Round 3

Reviewer 1 Report

The authors have addressed the concerns raised and improved the quality of their manuscript. Publication is recommended.